# Low rates of serious complications and further procedures following surgery for base of thumb osteoarthritis: analysis of a national cohort of 43 076 surgeries

Jennifer CE Lane [ID],[1] Richard Craig [ID],[1] Jonathan L Rees,[1] Matthew Gardiner [ID],[1,2] Mark M Mikhail [ID],[3] Nicholas Riley,[4] Daniel Prieto-Alhambra [ID],[1] Dominic Furniss[1,3,4]

DP-A and DF contributed equally.

For numbered affiliations see end of article.

**Correspondence to**
Prof Dominic Furniss;
dominic.furniss@ndorms.ox.ac.uk

## ABSTRACT

**Objectives** To determine the incidence of further procedures and serious adverse events (SAEs) requiring admission to hospital following elective surgery for base of thumb osteoarthritis (BTOA), and the patient factors associated with these outcomes.

**Design** Population based cohort study.

**Setting** National Health Service using the national Hospital Episode Statistics data set linked to mortality records over a 19-year period (01 April 1998–31 March 2017).

**Participants** 43 076 primary surgeries were followed longitudinally in secondary care until death or migration on 37 329 patients over 18 years of age.

**Main outcome measures** Incidence of further thumb base procedures (including revision surgery or intra-articular steroid injection) at any time postoperatively, and local wound complications and systemic events (myocardial infarction, stroke, respiratory tract infection, venous thromboembolic events, urinary tract infection or renal failure) within 30 and 90 days. To identify patient factors associated with outcome, Fine and Gray model regression analysis was used to adjust for the competing risk of mortality in addition to age, overall comorbidity and socioeconomic status.

**Results** Over the 19 years, there was an increasing trend in surgeries undertaken. The rate of further thumb base procedures after any surgery was 1.39%; the lowest rates after simple trapeziectomy (1.12%), the highest rates after arthroplasty (3.84%) and arthrodesis (3.5%). When matched for age, comorbidity and socioeconomic status, those undergoing arthroplasty and arthrodesis were 2.5 times more likely to undergo a further procedure (subHR 2.51 (95% CI 1.81 to 3.48) and 2.55 (1.91 to 3.40)) than those undergoing simple trapeziectomy. Overall complication rates following surgery were 0.22% for serious local complications and 0.58% for systemic events within 90 days of surgery.

**Conclusions** The number of patients proceeding to BTOA surgery has increased over the last 19 years, with a low rate of further thumb base procedures and SAEs after surgery overall registered. Arthrodesis and arthroplasty had a significantly higher revision rate.

## Strengths and limitations of this study

► To our knowledge, this is the largest scale national study identifying the risks following surgery for base of thumb osteoarthritis.

► Long follow-up is enabled by a nationalised health system linked to mortality records, that can identify surgical intervention in England over the last 19 years.

► Using data from a nationalised health system has the benefit of capturing data from patients who present to another hospital for treatment of their complication.

► As this study uses data collected from routine clinical care, results have a greater chance of being generalisable to a whole population.

► This data set can only register complications presenting to hospital, so is limited to identifying serious adverse events requiring day case admission or further surgery.

**Trial registration number** NCT03573765.

## INTRODUCTION

### Background

Osteoarthritis of the first carpometacarpal joint, or base of thumb osteoarthritis (BTOA), is very common and may need surgical treatment.[1] Worldwide, the majority of surgical interventions are simple trapeziectomy with or without ligament reconstruction and tendon interposition (LRTI).[1–3] However, there is increasing interest in the use of arthroplasty for BTOA, with many different implants available.[4–10] Concerns have been raised regarding increased risk of complications with LRTI and arthroplasty compared with simple trapeziectomy.[11 12] A Cochrane review comparing surgical techniques

concluded that there was a high risk of bias within current studies, with no single technique presenting superior results.[13] Recent work aimed to identify complications within a large insurance provider in the USA for small joint hand osteoarthritis procedures, and found complication rates as high as 35%, including 5% risk of a complication related to the prosthesis within 2 years of surgery.[14]

Many surgical studies in BTOA compare two techniques either within a clinical trial or single centre cohort, but there are very few studies comparing all available techniques with a long follow-up at a national level. Routinely collected data produced from everyday clinical practice offers the advantage of enabling long-term follow-up and detection of rare outcomes for healthcare interventions in the general population.[15] Routinely collected data for hospital admissions within the nationalised health service in England has been widely used to evaluate the safety and outcomes following surgery for other studies musculoskeletal conditions.[16–18]

The primary aim of this study was to estimate the incidence of BTOA surgery, further thumb base procedures and complications requiring admission to hospital in adults in the National Health Service (NHS) in England. Secondary aims were to identify factors associated with adverse outcomes, including registered local complications and systemic events, and further thumb base procedures after surgery. Our hypothesis was that there is variation in outcome between surgical subtypes.

## METHODS

### Data source

A pseudonymised, bespoke extract was made from the NHS database Hospital Episode Statistics Admitted Patient Care (HES APC) covering the period 1 April 1998 to 31 March 2017.[19] This individual patient level data extract contained all patient episodes of care within the NHS England system, including those episodes that occur within independent providers that are remunerated by the NHS. The NHS provides the majority of healthcare in England, with only 13% of elective surgery estimated to be privately funded.[20]

Episodes of care are linked via a patient's individual NHS number, therefore enabling linkage of all treatments undertaken and longitudinal follow-up, including complications and revision surgeries undertaken and registered by any provider. The HES APC extract was also linked to the Office for National Statistics (ONS) national mortality data set prior to pseudonymisation to identify cause and date of death.[21] By design, raw data was collected by NHS Digital prior to delivery to the research team.

### Population

All patients over 18 years undergoing BTOA surgery were identified using a list of OPCS Classification of Interventions and Procedures (OPCS V.4.8) to identify the procedures undertaken and International Classification of Diseases (ICD-10) codes to identify disease associated with the procedure (online supplemental tables 1–3).[22 23] OPCS codes are used in England to classify any procedure undertaken, and are used to identify a combination of the implant and surgical subtype used in association with the anatomical area of the body where the surgery took place. For hand surgery, OPCS codes are used with combination of a generic surgical procedure codes and anatomical location.

All episodes of care for each individual prior to and following BTOA surgery were included in the extract. Patients were followed up until date of death or the end of the study period (31 March 2017). Patients who had an ICD-10 code for traumatic injury in the same episode as BTOA surgery were excluded, in order to only include patients undergoing elective surgery for longstanding BTOA. Duplicate episodes can occur over the change of financial year, and were removed during data cleaning. Each hand was considered as a separate surgical case.

All cases of BTOA surgery identified within the 19-year period were included if they were not associated with a fracture ICD diagnosis code, and were not considered to be a duplicate episode. All cases were included with the aim of identifying the full national cohort within the time frame. Minimum follow-up was set at 1 day, due to the clinical opinion of the team that further procedure or serious adverse events (SAEs) can occur very shortly after surgery (eg, acute carpal tunnel syndrome requiring decompression).

Validation: Determining the feasibility of identifying surgical procedures and BTOA cases two validation studies were undertaken in order to determine if it was possible to successfully identify surgical treatment of BTOA in this data set, and second to determine if it was possible to identify the surgical subtypes undertaken.

First, we undertook consensus discussion between surgeons, coders and NHS Digital to determine the most likely OPCS procedure codes and ICD diagnosis codes that would identify surgery for BTOA within routinely collected hospital data in England. This was an iterative process within a diverse team of stakeholders. Following this, an external validation study was then undertaken using this list of OPCS and ICD codes that identified a positive predictive value of 81% for incident BTOA with good interobserver reliability.[24]

Second, we undertook a further external validation study to determine if OPCS procedure codes could identify surgery subtypes within routinely collected data in the NHS in England. This was undertaken in our NHS trust using two blinded and independent reviewers (MMM and NR), who reviewed a year of surgical activity and the OPCS and ICD codes used during this time. In a year's sample of 104 patients undergoing BTOA surgery in our institution, we demonstrated a positive predictive value of 99% in identifying surgical subtype.

## Covariates

Socioeconomic status of patients was identified using the Index of Multiple Deprivation (IMD), a government generated score of deprivation on geographical location in England.[25] Overall comorbidity level was calculated using the Charlson Comorbidity Index (CCI).[26 27] Ethnicity, as collected by NHS Digital, was included as a baseline covariate of interest, as national data is more inclusive than trial data, allowing us to identify any effect of ethnicity.[28] Baseline incidence of other conditions associated with BTOA pathogenesis in the literature were also identified using OPCS and ICD codes from previous episodes of secondary care, including a medical history of carpal tunnel syndrome, knee osteoarthritis, generalised osteoarthritis, rheumatoid arthritis, wrist fracture and oophorectomy.

## Further thumb base procedures

Further thumb base procedure was defined as any revision surgery or intra-articular steroid injection in the same thumb base following surgery. In order to identify factors associated with further thumb base procedure, time to further procedure and produce an incidence rate, only further procedures registered with BTOA surgery or injection codes *in the same hand* were used, identified through a laterality code. Also, to produce an estimation of a 'worst case' scenario of the percentage of cases that *may* have proceeded to further thumb base procedure, patients who had three or more episodes with BTOA surgery or injection codes but missing laterality were included in the percentage, but were not included in all subsequent analyses.

## Complications

Complications were grouped into local (surgical site infection or dehiscence requiring surgery, and neurovascular or tendon injury) and systemic events (stroke, respiratory tract infection, acute myocardial infarction, venous thromboembolic disease, urinary tract infection and acute renal failure). All complications had to be registered within HES APC to be identified and therefore required at least day case admission or surgery; minor complications will therefore not be registered in the data set. Registered complications were defined within 30 and 90 days of surgery to align with NHS outcomes framework used to compare outcomes between different medical conditions.[29] Furthermore, in the NHS, all surgical activity is compared with regards to outcomes within 30 and 90 days, and therefore exploring SAEs that occur within this time frame enables comparison to other musculoskeletal conditions and limb surgeries, and to studies already published in the literature.[16–18] It was also our clinical experience that these serious local and systemic complications requiring hospital admission were likely to present within these time periods. Registered serious systemic events were studied in addition to serious local complications due to concern about the side effects of general anaesthesia associated with upper limb procedures, and to enable comparison to rates registered following surgery for other musculoskeletal conditions.[17 18]

## Statistical methods

Age and sex specific incidence of surgery were calculated using official national mid-year population estimates in order to reflect the national nature of the healthcare system being studied, and to enable comparison to other musculoskeletal conditions, or study results from other countries.[30] Producing age and sex specific incidence also enables the results to take into account the change in population demographics over the long time period studied. All complications were calculated as a proportion of the sample with 95% CIs. Incomplete records were found for 0.95% of cases for age, sex, ethnicity and IMD deciles. Data was assumed to be missing at random, and a complete case analysis undertaken without imputation. Laterality was missing in 16.2% of cases, and in order to assess if missing laterality caused a bias in included cases, the baseline demographics of patients with and without laterality were compared (online supplemental table 4). No difference in baseline demographics was seen and therefore this was not considered as a potential cause of bias.

Factors associated with a further thumb base procedure were identified using a multivariable regression analysis using a Fine and Gray model that accounts for the competing risk of mortality.[31] The Fine and Gray model is a proportional hazards model that produces a subHR, that is derived from the cumulative incidence function (cause specific hazard). The Fine and Gray model splits the risk into two parts, the risk of further procedure, against (in this context) the risk of death. This is important within this study due to the long follow-up and potential for patients to die within the follow-up period, and allows the model to identify those who undergo a further thumb base procedure only within those who remain alive. Adjustment was undertaken for sex, age, socioeconomic status (using IMD) and overall comorbidity (using CCI). As registered complications were assessed within 90 days of surgery, multivariable cox proportional hazard modelling was used without adjusting for the competing risk of mortality. Patient age was identified as having a non-linear relationship with further thumb base procedure, and therefore was categorised, with the category containing the median age used as the reference category. When dividing the risk of further procedure by surgical subtype, there were insufficient patients below the age of 40 for inclusion in the multivariable regression analysis. Multivariable regression analysis was undertaken for all patient requiring further thumb base procedure, and also for those who had trapeziectomy alone, trapeziectomy with LRTI, arthroplasty and arthrodesis. For partial trapeziectomy there were insufficient numbers of patients who had a further thumb base procedure to undertake this analysis.

In order to assess the impact of surgical subtype, the main multivariable regression analysis was undertaken to compare the risk of further thumb base procedure for

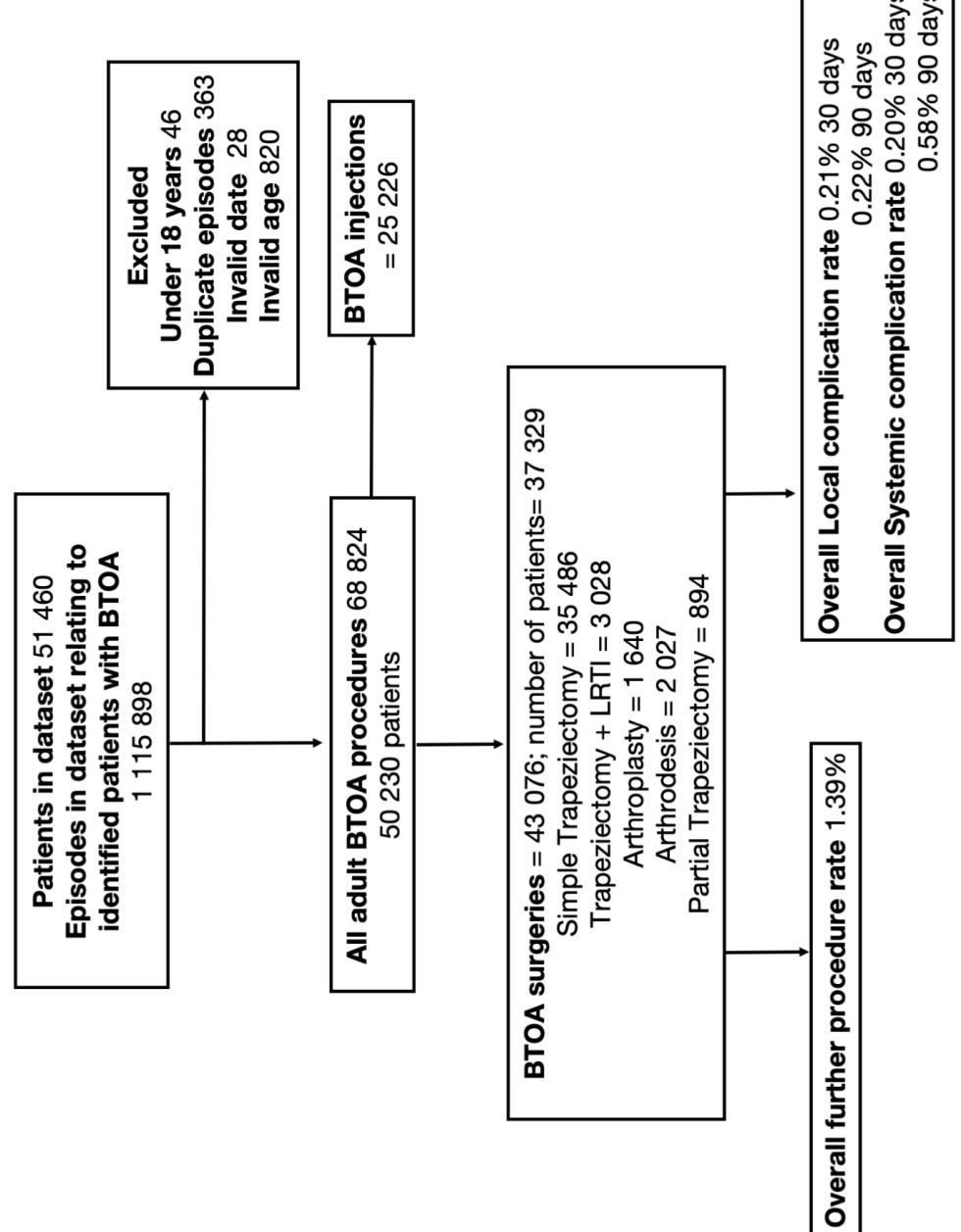

**Figure 1** Data processing flow chart. BTOA, base of thumb osteoarthritis; LRTI, ligament reconstruction and tendon interposition.

each surgical subtype compared with simple trapeziectomy. Further sensitivity analyses were undertaken, where each surgical subtype was assessed as an individual cohort in separate multivariable models adjusted for age, sex and socioeconomic status. These were undertaken to investigate each cohort of patients undergoing a particular surgical procedure, to assess for the potential of confounding by indication within the main model where all patients were combined as one group.

Statistical analysis was undertaken using Stata V.15.1. Significance within the multivariable regression analysis was determined for a subHR (sHR) when a 95% CI did not include 1, giving a more granular picture of the data than a p value alone. No power calculation was undertaken as

this study used all available data within a national cohort analysis.

**Patient and public involvement**

A James Lind Alliance priority setting partnership focused on common conditions affecting the hand and wrist also raised the question 'which interventions give the best results in the treatment of painful joints in the hand and wrist' as important to patients, making this study a direct response in its research question.[32] No patients were directly involved in the study design, conduct or interpretation of the study.

**Table 1** Demographics of patients undergoing base of thumb osteoarthritis surgery

| | All surgery (N, %) (total=43 076) | Trapeziectomy N (%) (35 486) | LRTI N (%) (3028) | Arthroplasty N (%) (1640) | Arthrodesis N (%) (2027) | Partial trapeziectomy N (%) (894) |
|---|---|---|---|---|---|---|
| Female sex | 34 112 (79.1) | 28 655 (80.8) | 2411 (79.6) | 1250 (76.2) | 1092 (53.9) | 704 (78.7) |
| Mean age (SD; years) | 63.2 (SD 9.2) | 63.6 (SD 8.8) | 62.5 (SD 9.7) | 61.7 (SD 9.3) | 58.1 (11.7) | 63.1 (9.8) |
| Charlson Comorbidity Index | | | | | | |
| 0 | 18 279 (42.4) | 14 992 (42.3) | 1324 (43.8) | 677 (41.3) | 890 (43.9) | 396 (44.3) |
| 1 | 10 640 (24.7) | 8807 (24.8) | 721 (23.8) | 444 (27.1) | 475 (23.4) | 193 (21.6) |
| 2 | 5693 (13.2) | 4746 (13.4) | 392 (13) | 199 (12.1) | 250 (12.3) | 106 (11.8) |
| 3 | 3286 (7.6) | 2723 (7.7) | 222 (7.3) | 125 (7.6) | 142 (7) | 74 (8.3) |
| 4 | 1659 (3.9) | 1359 (3.8) | 123 (4.1) | 60 (3.7) | 77 (3.8) | 40 (4.5) |
| >=5 | 3519 (8.2) | 2859 (8) | 246 (8.1) | 135 (8.2) | 193 (9.6) | 86 (8.5) |
| Index of Multiple Deprivation | | | | | | |
| Quintile 1 (least deprived) | 9518 (22.1) | 7949 (22.4) | 582 (19.2) | 435 (26.5) | 348 (17.2) | 204 (22.8) |
| Quintile 2 | 9637 (22.4) | 7950 (22.4) | 747 (24.6) | 342 (20.9) | 424 (21) | 174 (19.5) |
| Quintile 3 | 7961 (18.5) | 6553 (18.5) | 563 (18.6) | 264 (16.1) | 402 (19.8) | 179 (20) |
| Quintile 4 | 8004 (18.6) | 6550 (18.5) | 576 (19.1) | 309 (18.8) | 417 (20.6) | 152 (17) |
| Quintile 5 (Most deprived) | 7598 (17.7) | 6229 (17.5) | 514 (17) | 282 (17.2) | 393 (19.4) | 180 (20.1) |
| Missing | 358 (0.8) | 255 (0.7) | 46 (1.5) | 8 (0.5) | 43 (2) | 6 (0.7) |

LRTI, ligament reconstruction and tendon interposition .

## RESULTS

### Data processing

Figure 1 describes the data processing details. In the study period, 43 076 primary BTOA surgeries were performed on 37 329 individuals in English NHS hospitals; 79.1% were performed on female patients, and the mean age at primary surgery was 63 (SD 9.2). The median follow-up time was 1835 days (IQR 816–3223). Only 345 cases (0.8%) had a follow-up time of less than 30 days, and 1075 cases (2.5%) had a follow-up time of less than 90 days. It was therefore considered that loss to follow-up was not a sufficient source of bias within the cohort.

Demographics of patients undergoing BTOA surgery are shown in table 1, with further granularity given in online supplemental table 5. Over 60% of patients had no or very low levels of overall comorbidity, and 86% of patients were of white ethnic background. Patients were more likely to come from affluent sections of society. Approximately one in five patients had another hospital episode prior to BTOA surgery for osteoarthritis elsewhere in the body.

The most commonly undertaken surgical procedure was simple trapeziectomy, followed by trapeziectomy with LRTI. When comparing the baseline demographics of patients undergoing the different surgical subtypes, more men underwent arthrodesis and were slightly younger compared with the other groups. There appeared to be no difference in overall comorbidity as shown by Charlson index, and comparable socioeconomic status of patients shown by the IMD.

### Trends in surgery

Over the 19-year period there was an increasing trend in the number of BTOA surgeries undertaken, predominantly accounted for by a steady increase in simple trapeziectomy, with minimal change in the incidence of other surgical procedures (figure 2).

### Further thumb base procedures

The overall rate of further thumb base procedure after any BTOA surgery was 1.39%, with the median time to intervention being under 1.5 years (table 2; figure 3). While the rate of further thumb base procedures was low in general in this population, the highest rates were seen after arthroplasty (3.84%) and arthrodesis (3.50%) and the lowest following simple trapeziectomy (1.12%). The survival of BTOA surgery over time according to surgical subtype is given in Kaplan-Meier plots in the appendix (online supplemental figures 1–5), noting the majority of patients proceeded to further intervention around the first 18 months postoperatively.

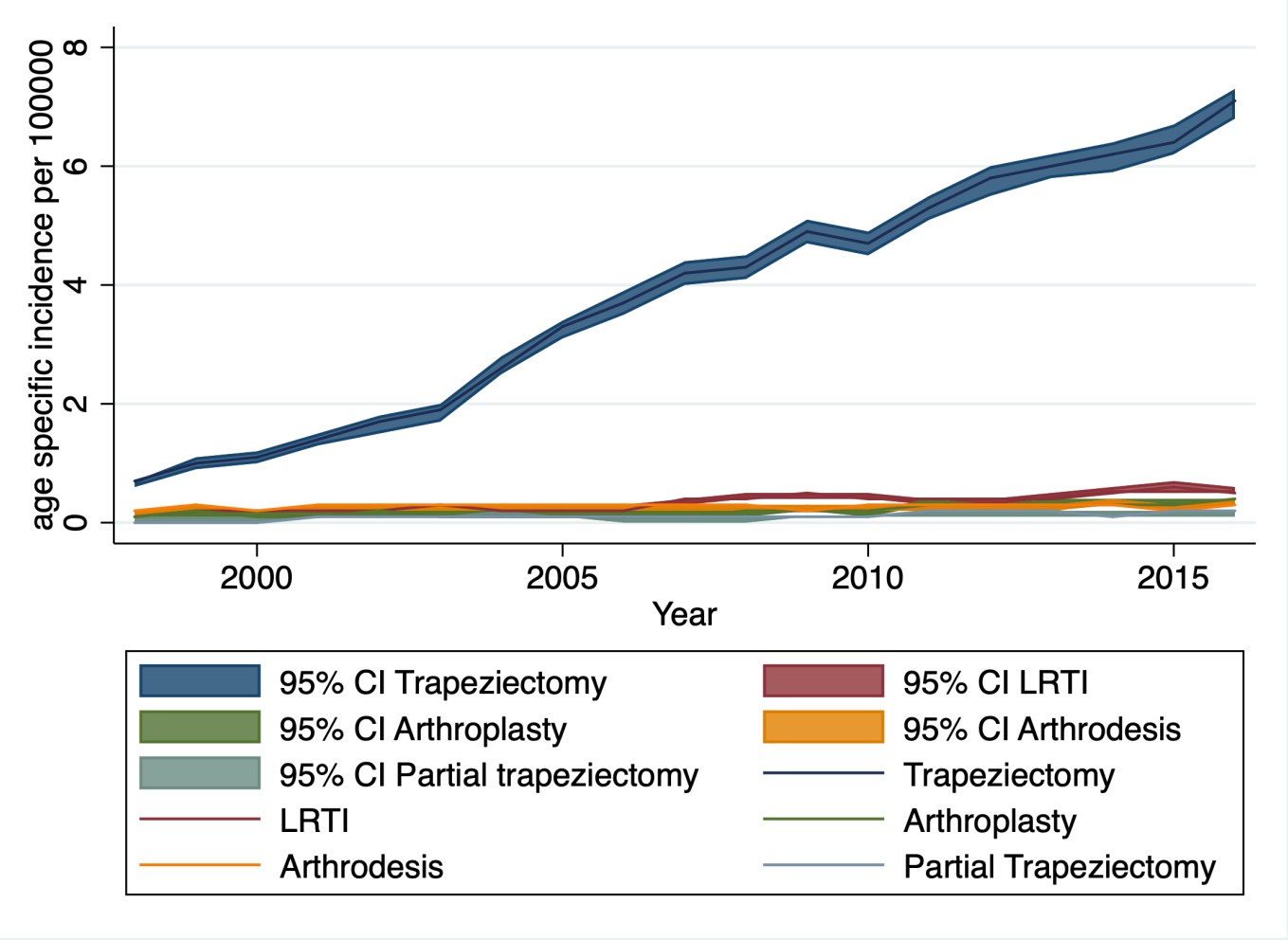

**Figure 2** Age-specific incidence of base of thumb osteoarthritis surgeries by year, England 1998–2017. LRTI, ligament reconstruction and tendon interposition.

### Complications

Overall there was a low rate of registered serious local complications and systemic events requiring admission to hospital following surgery, with the risk of any serious local complication being 0.22% within 90 days, and 0.58% for systemic events within 90 days (table 3 and online supplemental table 6). There was an insufficient number of cases to enable further analysis of factors associated with complications.

### Factors associated with further thumb base procedure

When considering the impact of age, sex and sociodemographic factors, the risk of further thumb base procedure was increased for those in a younger age category of 40–49 years (adjusted sHR 1.53 (95% CI 1.10 to 2.13)) and for men (adjusted sHR 1.24 (95% CI 1.01 to 1.53)) (online supplemental table 7) when adjusting for the other factors. When adjusting for surgical subtype in addition to age, sex and socioeconomic status to determine if surgical subtype had an impact on the risk of further procedure, the increased risk for men and those who underwent surgery at a younger age did not remain significant. This suggests that increased risk of further

thumb base procedure was likely to be due to a greater proportion of younger male patients undergoing arthroplasty and arthrodesis (online supplemental table 8).

When adding in surgical subtype as a variable of interest into the main analysis, compared with those undergoing simple trapeziectomy, those undergoing arthroplasty and arthrodesis were around 2.5 times more likely to proceed to further thumb base procedure (arthroplasty 2.46 (1.77 to 3.42); arthrodesis 2.41 (1.78 to 3.28)) after adjustment for age, sex and socioeconomic deprivation. The forest plot in figure 4 shows the impact of age, sex, socioeconomic deprivation and surgical subtype seen on the risk of undergoing a further procedure.

In the sensitivity analyses where each surgical subtype was run in a separate model adjusted for age, sex and deprivation, for the risk of further thumb base procedure within each surgical subtype, an increased risk of further procedure was seen in men for those who underwent trapeziectomy alone when adjusted for age and deprivation (adjusted sHR 1.33 (95% CI 1.02 to 1.73)). This was not seen in those undergoing LRTI, arthrodesis or arthroplasty, but noting there were fewer cases in these

**Table 2** Rate of further thumb base procedure (revision surgery or intra-articular steroid injection) according to BTOA surgical subtype

| | Number of cases | Median follow-up time in days (IQR) | Number cases needing further procedures* (%) | Median time to further procedure* in days (IQR) | Further procedure* rate per 1000 person years (95% CI) |
|---|---|---|---|---|---|
| All BTOA surgeries | 43 076 | 1835 (816 to 3223) | 599 (1.39) | 472 (272 to 965) | 1.72 (1.54 to 1.92) |
| Trapeziectomy | 35 486 | 1839 (820 to 3228) | 398 (1.12) | 420.5 (248 to 804) | 1.73 (1.55 to 1.93) |
| Trapeziectomy with LRTI | 3028 | 2063.5 (880 to 3405) | 45 (1.49) | 508 (287 to 1648) | 2.13 (1.56 to 2.92) |
| Arthroplasty | 1640 | 2000.5 (865 to 3683.5) | 63 (3.84) | 732.5 (325 to 1666.5) | 4.83 (3.65 to 6.40) |
| Arthrodesis | 2027 | 2712.5 (1222 to 4546) | 71 (3.50) | 520 (301 to 1054) | 3.91 (3.04 to 5.04) |
| Partial trapeziectomy | 894 | 1972 (929 to 4070) | 22 (2.46) | 673.5 (296.5 to 1599.5) | 2.63 (1.59 to 4.37) |

*Further thumb base procedure only (intra-articular steroid injection or revision surgery).
BTOA, base of thumb osteoarthritis; LRTI, ligament reconstruction and tendon interposition .

subgroups. Age and sociodemographic status did not appear significant in any of the surgical subgroups in regression analysis. (online supplemental tables 9–12).

## DISCUSSION
### Principle findings
This study found that patients undergoing surgery for BTOA in the NHS in England are predominantly women

and in their seventh decade. There has been an increase in surgery undertaken over the last 19 years. There was a low rate of revision surgery and intra-articular steroid injection registered following primary surgery in this cohort. In addition, there are low rates of registered serious local complications and systemic events requiring hospital admission within the immediate postoperative period. Further thumb base procedures mostly occurred

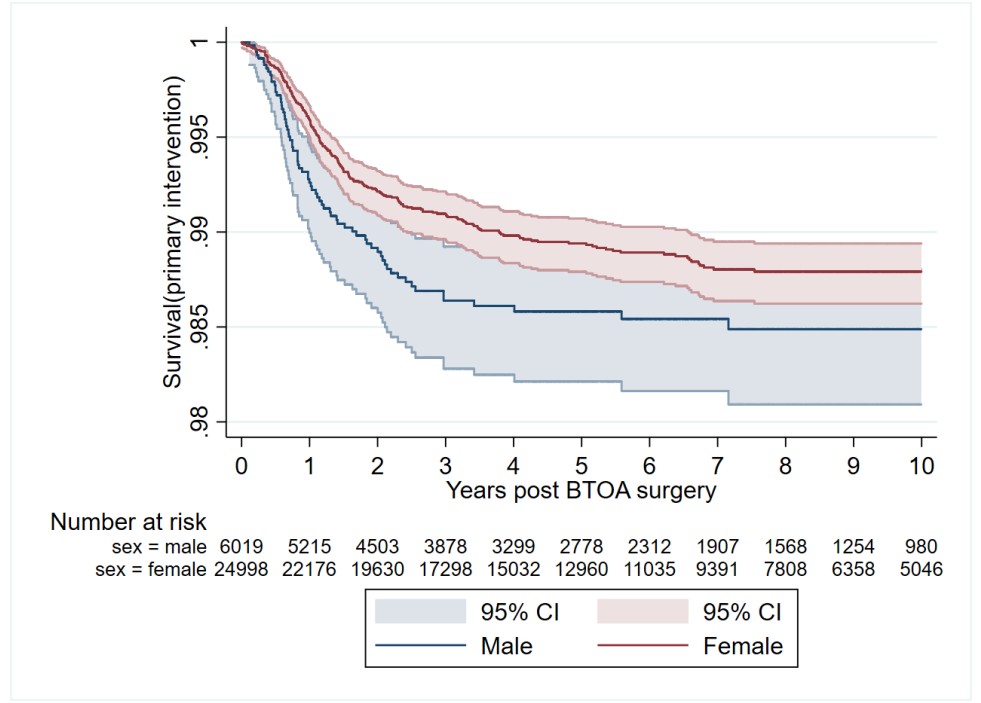

**Figure 3** Kaplan-Meier plot of risk of further thumb base procedure after all BTOA surgery, divided by sex. BTOA, base of thumb osteoarthritis.

**Table 3** Local complications and systemic events following all base of thumb osteoarthritis surgery

| | Time | All surgery N (% (95% CI)) |
|---|---|---|
| **Local complications** | | |
| Wound dehiscence and wound infection | Within 30 days | 12 (0.03 (0.01 to 0.05)) |
| | Within 90 days | 14 (0.03 (0.02 to 0.05)) |
| Neurovascular injury | Within 30 days | 79 (0.18 (0.15 to 0.23)) |
| Any complication | Within 30 days | 91 (0.21 (0.17 to 0.26) |
| | Within 90 days | 93 (0.22 (0.17 to 0.26) |
| **Systemic events** | | |
| Stroke | Within 30 days | 9 (0.02 (0.01 to 0.04)) |
| | Within 90 days | 40 (0.09 (0.07 to 0.13)) |
| Respiratory tract infection | Within 30 days | 70 (0.16 (0.13 to 0.21)) |
| | Within 90 days | 174 (0.40 (0.35 to 0.47)) |
| Myocardial infarction | Within 30 days | 17 (0.04 (0.02 to 0.06)) |
| | Within 90 days | 50 (0.11 (0.09 to 0.15)) |
| DVT/PE | Within 30 days | 18 (0.04 (0.02 to 0.07)) |
| | Within 90 days | 41 (0.10 (0.07 to 0.13)) |
| Urinary tract infection (UTI) | Within 30 days | 30 (0.07 (0.05 to 0.10) |
| | Within 90 days | 75 (0.17 (0.14 to 0.22)) |
| Acute renal failure (ARF) | Within 30 days | 13 (0.03 (0.02 to 0.05)) |
| | Within 90 days | 44 (0.10 (0.07 to 0.14)) |
| Any systemic event | Within 30 days | 87 (0.20 (0.16 to 0.25)) |
| | Within 90 days | 250 (0.58 (0.51 to 0.66)) |
| Any systemic event (excluding UTI and ARF) | Within 30 days | 44 (0.10 (0.07 to 0.14)) |
| | Within 90 days | 131 (0.30 (0.25 to 0.36)) |

DVT/PE, Deep vein thrombosis / pulmonary embolism.

within the first 18 months postoperatively. The greatest factor influencing progression to further thumb base procedure in this population was surgical subtype, with patients receiving an arthroplasty around 2.5 times more likely to undergo further procedure than those having simple trapeziectomy.

### Strengths and contribution to current knowledge

The major strength of this study is the national nature of the study setting. The HES APC data set has been extensively used for evaluating the safety and outcomes following surgery for other musculoskeletal conditions, and this large national cohort of patients provides longitudinal follow-up of individuals across multiple providers.[16–18] It allows retention in the study of those individuals who may have sought a second opinion at a different hospital in England. This gives a better overview of the need for revision surgery and for the management of SAEs following surgery than is possible in single centre or regional studies. As the private healthcare sector accounts for a low proportion of all healthcare activity in England, while some may seek consultation in this sector, all NHS activity will be captured within this data set.[20] It also accounts for patient movement within the country,

noting that emigration out of the UK is thought to be low at around 400 000 persons per year.[33]

This study includes all adult patients who underwent surgery in England, and is representative of the generality of hand surgery practice in a large public healthcare system. Patients at extremes of ages, with comorbid conditions that may render them ineligible to take part in clinical trials, are included. Due to practical and financial constraints, randomised control trials typically lack the statistical power and longevity of follow-up to reliably inform on the safety of surgical procedures for BTOA. Large observational studies, such as this one, are important to fill that evidence gap. Previous studies have suggested increased revision risk in those of a younger age which we also detected here in a univariable analysis. The effect did not persist here in multivariable analysis, and may therefore be more directly related to choice of surgery type.[34 35]

All widely used surgical procedure subtypes are included in this study, enabling a full comparison of each main procedure with the other. This facilitates broad comparison of the rate of registered complications and revision across an inclusive population with longer

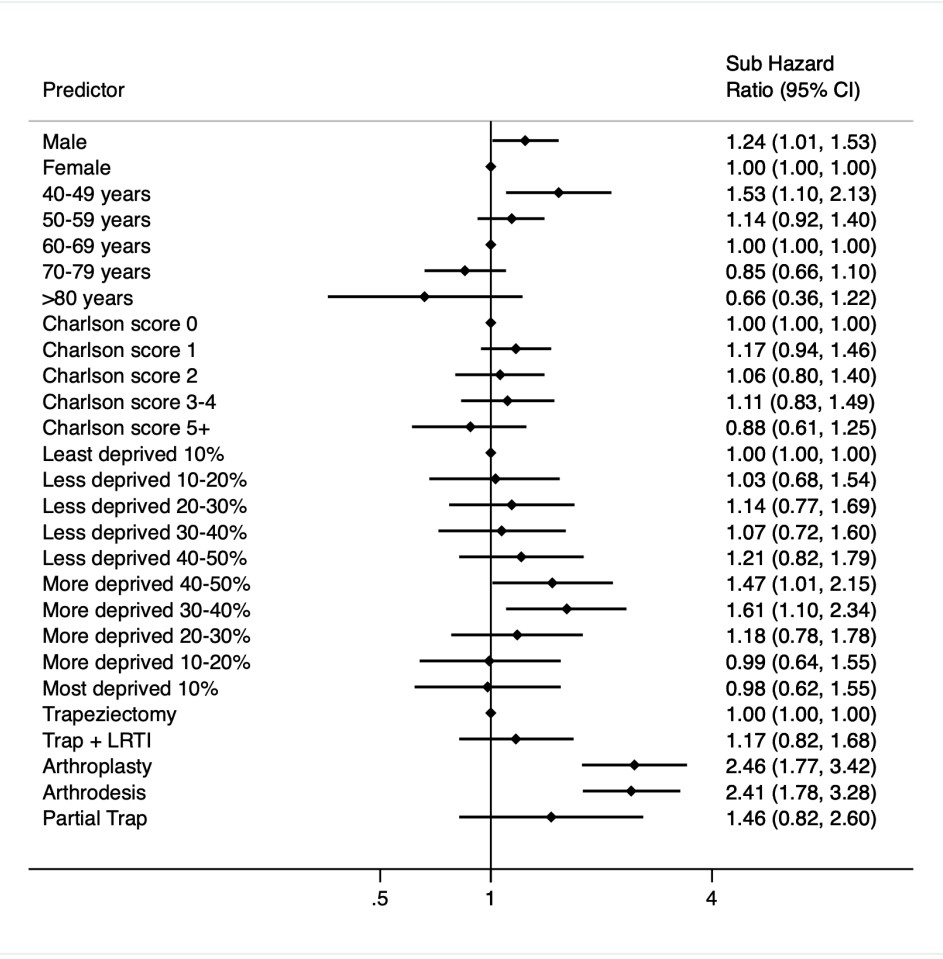

| Predictor | | Sub Hazard Ratio (95% CI) |
|---|---|---|
| Male | | 1.24 (1.01, 1.53) |
| Female | | 1.00 (1.00, 1.00) |
| 40-49 years | | 1.53 (1.10, 2.13) |
| 50-59 years | | 1.14 (0.92, 1.40) |
| 60-69 years | | 1.00 (1.00, 1.00) |
| 70-79 years | | 0.85 (0.66, 1.10) |
| >80 years | | 0.66 (0.36, 1.22) |
| Charlson score 0 | | 1.00 (1.00, 1.00) |
| Charlson score 1 | | 1.17 (0.94, 1.46) |
| Charlson score 2 | | 1.06 (0.80, 1.40) |
| Charlson score 3-4 | | 1.11 (0.83, 1.49) |
| Charlson score 5+ | | 0.88 (0.61, 1.25) |
| Least deprived 10% | | 1.00 (1.00, 1.00) |
| Less deprived 10-20% | | 1.03 (0.68, 1.54) |
| Less deprived 20-30% | | 1.14 (0.77, 1.69) |
| Less deprived 30-40% | | 1.07 (0.72, 1.60) |
| Less deprived 40-50% | | 1.21 (0.82, 1.79) |
| More deprived 40-50% | | 1.47 (1.01, 2.15) |
| More deprived 30-40% | | 1.61 (1.10, 2.34) |
| More deprived 20-30% | | 1.18 (0.78, 1.78) |
| More deprived 10-20% | | 0.99 (0.64, 1.55) |
| Most deprived 10% | | 0.98 (0.62, 1.55) |
| Trapeziectomy | | 1.00 (1.00, 1.00) |
| Trap + LRTI | | 1.17 (0.82, 1.68) |
| Arthroplasty | | 2.46 (1.77, 3.42) |
| Arthrodesis | | 2.41 (1.78, 3.28) |
| Partial Trap | | 1.46 (0.82, 2.60) |

**Figure 4** Forest plot of relative risk of further thumb base procedure after primary base of thumb osteoarthritis surgery within multivariable regression analysis adjusted for age, sex, sociodemographic status and surgical subtype. LRTI, ligament reconstruction and tendon interposition.

follow-up than most clinical trials.[12 13] This study emphasises that while the overall risk of BTOA surgery is low, there is an increased risk of further procedure for those undergoing arthroplasty that warrants further large scale epidemiological analysis, as has been seen in joint arthroplasty research at other anatomical sites.[36 37]

### Limitations and future work

This study focusses on the patient factors that can impact the risk of adverse outcome. It does not account for potential variation associated with surgeon or hospital factors. Further methodological research is underway to identify the best method of multilevel analysis that can explore the interaction of healthcare provider factors with patient factors. This technique may enable us to identify how multiple factors influence the risk of adverse events.

Patients included here are those who have been registered within NHS Digital records, a data set that is designed for remuneration rather than research. This means therefore that minor complications that do not require a minimum of a day case admission or procedure will not be recorded. This study can identify the most serious complications and events that impact on morbidity, but study of minor complications are better

suited to electronic healthcare record or primary care data analysis.

Care was taken to validate the codes in two different populations used to identify patient records in order to limit misclassification bias, but this remains a potential limitation of the work. All arthroplasties are coded as a generic arthroplasty code, and therefore implant type cannot be analysed. Future work focussing on implant registries, such as currently used in hip and knee arthroplasty, is needed to determine the impact of implant type on outcome in routine clinical practice.[38 39] Members of our department recently undertook a service evaluation project in 15 centres in the UK.[40] In this study of 150 patients, there was a 50:50 split between LRTI and simple trapeziectomy undertaken. However, this was undertaken in centres volunteering to participate in research, with data collected by surgeons, with a much smaller sample size and over a much shorter period of time (March 2017–May 2019). This study may therefore be prone to selection bias in a different way to the selection bias generated by using routinely collected data. In order to determine if we could adequately identify surgical subtypes from the HES APC data set, we undertook two external validation studies

after an iterative process of generating an included list of diagnosis and procedure codes. After these processes, we demonstrated that it was possible to differentiate between the surgical subtypes, including simple trapeziectomy and LRTI. Our data suggest that simple trapeziectomy was undertaken much more commonly than LRTI in England. Randomised control trials based in the UK and a Cochrane review have suggested little difference in efficacy between LRTI and simple trapeziectomy, and higher costs and complications associated with LRTI, and therefore the low proportion of LRTIs undertaken could reflect English surgeons adhering to UK evidence based practice.[2 11 13]

Comorbidities registered in this data set are necessarily registered within an admitted patient episode, which may lead to inclusion bias of information used for remuneration. As the registration of comorbidities is likely to reflect the selection bias of only those who have received inpatient care for these conditions, these conditions were not used as factors that may influence the need for further thumb base procedures within the multivariable analysis. While admitted patient care gives the best granularity of information for surgery and in-hospital events, it does not cover interactions occurring in an outpatient or primary care environment. Therefore, this study only identifies SAEs; those requiring at least a day case admission to hospital, and will not include those complication seen in primary care or outpatient facilities. It will fail to identify some pre-existing comorbidities in our patients, or persistent pain. Complications treated in the community such as minor surgical site infections requiring oral antibiotics will also not be registered within this data set. Our future work is focused on identifying patients who have undergone BTOA surgery in primary care records, to compare the information collected there regarding comorbidity and postoperative infection.

Finally, this study uses revision surgery and SAEs as a proxy for outcome following surgery, as no patient-reported outcome measures (PROMs) are routinely collected for hand surgery at a national level by the NHS. Further work is needed to focus on comparing PROMs for each surgical subtype in routine clinical practice to identify the quality of outcome from the patient perspective.

## Meaning and use for clinical practice

As this study includes a national cohort of patients from routine clinical practice, the rates of further thumb base procedure and SAEs requiring inpatient hospital treatment given in this study are directly of use for counselling patients during the process of consent and shared decision-making. The higher rates of further thumb base procedure following arthroplasty found in this study concurs with a previous systematic review of smaller trials.[12] Within a large cohort of patients, demographic factors did not appear to be associated with significantly increased risk of adverse outcomes when accounting for surgical subtype, which again is informative for both the surgeon and patient. The study suggests that there is a low overall rate of further procedures and serious complications following surgery for BTOA. While the risk of further procedure does vary by surgical subtype, demographic factors do not appear to influence this risk.

**Author affiliations**
[1]Nuffield Department of Orthopaedics, Rheumatology and Musculoskeletal Sciences, Oxford University, Oxford, UK
[2]Department of Plastic Surgery, Wexham Park Hospital, Slough, UK
[3]Department of Plastic Surgery, Oxford University Hospitals NHS Foundation Trust, Nuffield Orthopaedic Centre, Oxford, UK
[4]Department of Hand Surgery, Oxford University Hospitals NHS Foundation Trust, Nuffield Orthopaedic Centre, Oxford, UK

**Acknowledgements** The authors would like to acknowledge Miss Rachel Kuo and Professor Jane Green for their work in this study.

**Contributors** JCEL, DF, RC, JLR, MG and DP-A conceived the study and designed the study method. MMM and NR undertook the validation study. JCEL and RC acquired the data. JCEL performed the data analysis. JCEL, DF, RC, JLR, MG, NR and DP-A interpreted the data. JCEL drafted the manuscript. All authors reviewed and edited the final manuscript. JCEL, RC, JLR and DF has full access to all data in the study. DF is the guarantor.

**Funding** This work was supported by Versus Arthritis (21605) (JLR); the Medical Research Council (MR/K501256/1) (JLR); RCS England/NJR research fellowship (RC); University of Oxford (JLR); the National Institute for Health Research (NIHR) (SRF-2018-11-ST2-004) (DP-A); NIHR Biomedical Research Centre, Oxford (BRC) (RC, DF and JLR) and Oxford Medical Research Fund (MG). No funders had a direct role in this study. The views and opinions expressed are those of the authors and do not necessarily reflect those of the Clinician Scientist Award programme, NIHR, NHS or the Department of Health. This work used data supplied by NHS Digital copyright 2018, re-used with the permission of NHS Digital. All rights reserved.

**Competing interests** All authors have completed an ICJME conflict of interest form that is uploaded with the study (http://www.icmje.org/conflicts-ofinterest/) no support from any organisation for the submitted work; DP-A has received research grants from Amgen, Servier, UCB; departmental fees for speaker services from Amgen, departmental fees for consultancy from UCB. JLR reports grants from the Medical Research Council (MR/K501256/1) and Versus Arthritis (21605), outside of the submitted work. NR reports personal fees from Acumed, outside the submitted work; Dr Craig reports grants from Royal College of Surgeons/National Joint Registry, outside the submitted work. No other relationships or activities could appear to have influenced the submitted work.

**Patient consent for publication** Not required.

**Ethics approval** This study was registered at ClinicalTrials.gov (NCT03573765) and approved by the University Research Services Clinical Trials Research Group (project ID 12787), and the NHS Data Access Advisory Group (DAAG). It was carried out in accordance with the NHS Digital Data Sharing Agreement (DARS-NIC-29827-Q8Z7Q). Studies using non-identifiable records from Hospital Episode Statistics are exempt from research ethics committee approval. Patients have the right to request that their data is not released by NHS Digital for use by researchers (register a 'Type 2 opt-out').

**Provenance and peer review** Not commissioned; externally peer reviewed.

**Data availability statement** Data may be obtained from a third party and are not publicly available. No further data can be made available due to NHS Digital restrictions. Data extracts can be applied for via the NHS Digital data access request service (https://digital.nhs.uk/services/data-access-request-service-dars).

**ORCID iDs**

Jennifer CE Lane http://orcid.org/0000-0002-1729-8654
Richard Craig http://orcid.org/0000-0003-0792-9288
Matthew Gardiner http://orcid.org/0000-0002-8058-4186
Mark M Mikhail http://orcid.org/0000-0001-5474-4193
Daniel Prieto-Alhambra http://orcid.org/0000-0002-3950-6346

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
