## [Reviewer comments · BMJ Open]

ARTICLE DETAILS

TITLE (PROVISIONAL)	Low rates of serious complications and further procedures following surgery for Base of Thumb Osteoarthritis: analysis of a national cohort of 43 076 surgeries
AUTHORS	Lane, Jennifer; Craig, Richard; Rees, J L; Gardiner, Matthew; Mikhail, Mark; Riley, Nicholas; Prieto-Alhambra, Daniel; Furniss, Dominic

VERSION 1 – REVIEW

REVIEWER	Tsujii, Masaya Mie University
REVIEW RETURNED	02-Nov-2020

GENERAL COMMENTS	Dear Authors, thank you for your manuscript. I'm very interested in your work. For the thumb carpometacarpal osteoarthritis, many surgical techniques have been reported even though high quality research is lacking. The authors evaluated the surgical techniques using big data of National Health Service in England. We could find few studies including a number of patients in surgical techniques for thumb carpometacarpal osteoarthritis. Therefore, this research showed very important data including trends of the techniques, complication, and revision rate. Nonetheless, have several comments and questions before publication of the journal. I understood that the data of NHS covered most of patients undergoing treatment for the diseases. I think that there must exist many researches for various diseases using the big data. Please mention the other researches in Introduction and Discussion. If you can find no research, please discuss the reason why the same method of this research was not used. Arthrodesis had more proportion of male than that of female. Did the authors perform statistical analysis in difference of the gender.? In addition, the authors should discuss the difference of the gender between the arthrodesis and the other techniques. In this study, simple trapeziectomy was predominantly more than the other techniques. I wonder the outcome. I understand that LRTI was almost equal to the simple trapeziectomy such as a recent study from U.K. (Parker S et al. The bone & joint journal. 2020). I cannot believe that the simple trapeziectomy was 10 times more than LRTI. How are the surgical techniques recorded in NHS of England? In my country, the differences could not be distinguished between the simple trapeziectomy and LRTI, while the arthrodesis and arthroplasty using artificial joint could be clearly found. If the data is correct, the authors need to describe
--

	the reason why this report showed more patients undergoing the simple trapeziectomy than previous researches. In abstract, the authors concluded arthrodesis and arthroplasty had a significantly higher revision rate. Did the authors performed statistical analysis for comparison of revision rate among the surgical techniques? In addition, which techniques was compared with arthrodesis and arthroplasty? Simple trapeziectomy? LRT1? Please discuss the increase of the simple trapeziectomy.
--	---

REVIEWER	Wouters, Robbert Erasmus Universiteit Rotterdam
REVIEW RETURNED	04-Nov-2020

GENERAL COMMENTS	In this study, the authors investigate the incidence of CMC-1 OA surgery and accompanied complications and additional procedures. For this purpose, the authors studied a large sample >37.000 individuals using routinely collected NHS data. The authors conclude that the number of CMC-1 OA surgeries has increased and that very few complications occurred (<1%) and few additional procedures were performed ($\pm 1.4\%$). Also, they conclude that revision rates were higher in patients with arthrodesis or arthroplasty. I think this is an interesting study, with, especially, a valuable large dataset of patients that underwent CMC-1 surgery. To my opinion, this has a lot of potential, but I have some serious concerns regarding the study in its present form. Main concerns: * My main concern is the reliability and representativeness of the registration of complications and additional procedures using OPCS and ICD codes. Ergo, is the reported number of complications and additional procedures reliable? My concern is that there may be a high risk of under reporting of complications, as we know that this is a frequent problem in studies on complications. Also, it may be that complications were not registered as requiring a new treatment episode. The reported numbers of complications seem unlikely low: e.g., in sup table 6 the numbers per surgery type are so low that it may be unrealistic. The numbers are much lower compared to other studies as well (e.g., Vermeulen et al. 2014 reports 25-37% complications). My doubts are strengthened by the impression that the reported numbers in sup table 4 on comorbidity (e.g., CTS, Knee OA, etc.) are low, especially since the population of CMC-1 OA has a higher prevalence compared to their peers of similar age. It is unclear how the authors addressed this. Hence, I think there is a high risk of bias here and the authors cannot safely study complication or additional procedure rate using this data, nor draw this conclusion. My advise would be to only study the prediction of these events and not imply that these rates are reliable. * In line with this, the classification of complications seems not really suitable. E.g., tendinitis, scar tenderness, temporary sensory loss, neuromas, thumb collapse etc., are not included here, whereas they should be considered a complication. Also, the time frame of 30 and 90 days is relatively short for these kind of joint procedures. Furthermore, complications that are considered systemic may not be correlated with the initial treatment, as, fore example, stroke seems not related.
--

	* As a non-UK person, I am not familiar with the OPCS system. For non-UK people, it would be helpful to explain this in more detail. * It seems unclear how the study sample and the follow-up period was arrived at. For example, the authors report median follow-up with IQR, but minimum follow-up would be very relevant as well. The title of this paper can be misleading as the median follow-up is much lower than 19 years (as for the subgroups median follow-up is 5-7 y). Also, inclusion criteria can be described more clearly. Currently, the authors only mention that patients with trauma and duplicates were excluded. Minor issues * The described validation process is not really clear to me, e.g., page 6 lines 6-12: what is meant by determining "the most likely OPCS procedure codes and ICD diagnosis codes"? Why was there a "validation study to identify BTOA as a condition" Why was this even necessary at all? I think that many readers will not be familiar with the registration system, and thereby will not understand the level of detail available (e.g., surgical subtype) and associated limitations. I think more explanation is necessary here. * The term "further procedure" may not be suitable, as this only covers revision surgery or intra-articular steroid injection in the same thumb base. I would suggest the term additional thumb base procedure * In the statistics section (page 7 lines 10-16), the authors describe the use of 95% CI's to determine significance in light of multiple testing. This is unclear and does not seem logical to me in preventing a multiple testing problem. Please explain in more detail. Also, I think many readers will not be familiar with subhazard ratios, the authors may consider to explain this in more detail as well. * The classification of ethnicity seems inconsistent, and I doubt the need of reporting this. * There is no referral to figure 2, I think this should be in the part on increasing trend of surgeries. Also, the description of this figure is quite unclear regarding the age-specific incidence. * It seems illogical to include separate models per surgical procedure when there is also a model with surgical procedure as a factor. This may be a bit overwhelming. * In the discussion, the authors conclude that surgery for CMC-1 OA is safe. I am not sure if that would be my conclusion based on this study. Also, Page 9 lines 48-51 and page 10 lines 3-6: see my main concern; I do not think that this study provides an accurate view of the number of complications and additional procedures due a high risk of bias in registration.
--	--

VERSION 1 – AUTHOR RESPONSE

Point by point responses

Reviewer: 1

Reviewer comments	Author response	Changes made	Page number in revised document

Dear Authors, thank you for your manuscript. I'm very interested in your work. For the thumb carpometacarpal osteoarthritis, many surgical techniques have been reported even though high quality research is lacking. The authors evaluated the surgical techniques using big data of National Health Service in England. We could find few studies including a number of patients in surgical techniques for thumb carpometacarpal osteoarthritis. Therefore, this research showed very important data including trends of the techniques, complication, and revision rate.	Thank you	NA	NA
1. I understood that the data of NHS covered most of patients undergoing treatment for the diseases. I think that there must exist many researches for various diseases using the big data. Please mention the other researches in Introduction and Discussion. If you can find no research, please discuss the reason why the same method of this research was not used.	You are correct- the HES APC dataset for hospital admissions in England has been widely used to investigate outcomes for other musculoskeletal diseases. Further description of the use of HES elsewhere has been added to clarify this	Addition to background & discussion	Addition to paragraph 2, page 4 and paragraph 2 page 9
2.Arthrodesis had more proportion of	Apologies that there was an omission in the statistical	Addition to statistical	Addition of word sex to paragraph 2 page

male than that of female. Did the authors perform statistical analysis in difference of the gender.? In addition, the authors should discuss the difference of the gender between the arthrodesis and the other techniques.	methods section- the multivariable regression was also adjusted for gender (sex), and therefore all the results of the multivariable model in the appendix show the impact of gender on the incidence of revision. Supplementary table 7 shows the impact of gender for the full model, with supplementary tables 9-12 showing the impact of gender when separated within surgical subtype. New paragraphs have been added to results and further clarification has been added to emphasise the impact of gender in addition to emphasising the greater detail available in the appendix.	methods section and to results	6, new paragraph 3 page 6, paragraph 2 page 7 and amendments paragraphs 1 and 2 page 9
3. In this study, simple trapeziectomy was predominantly more than the other techniques. I wonder the outcome. I understand that LRTI was almost equal to the simple trapeziectomy such as a recent study from U.K. (Parker S et al. The bone & joint journal. 2020). I cannot believe that the simple trapeziectomy was 10 times more than LRTI. How are the surgical techniques recorded in NHS of England? In my country, the differences could not be distinguished between the simple trapeziectomy and LRTI, while the arthrodesis and arthroplasty using	The work undertaken by Parker et al, members of our Department, was an audit of centres who volunteered to participate- the HES APC dataset contains all surgery undertaken in England that is remunerated by the NHS and therefore Parker provides a snapshot of surgery in just 15 units, and is prone to selection bias. Previous RCTs and a Cochrane review based in the UK have shown no difference in effectiveness between LRTI and simple trapeziectomy (with LRTI being more costly and with higher complications) and therefore we believe that in this study of all national practice, it is not inconceivable that simple trapeziectomy predominates over LRTI. In order to determine if it was possible to identify LRTI separately from trapeziectomy we undertook a validation study that showed we were able to differentiate with a positive	Clarification of validation study in methods to show steps taken in order to identify if coding was accurate, and further elaboration in the limitations section (including comparison to Parker et al), last paragraph on page 9	Methods section, new section called 'Validation: identifying surgical subtypes and BTOA cases' page 5, and limitations section, new second paragraph on page 9

artificial joint could be clearly found. If the data is correct, the authors need to describe the reason why this report showed more patients undergoing the simple trapeziectomy than previous researches.	predictive value of 99%. There is a potential limitation of inaccurate coding of LRTI, but since this comes with a higher tariff, it would be in the hospital's interest to code it appropriately to get adequate payment for surgery. We appreciate these points and have attempted to clarify them in the text.		
4. In abstract, the authors concluded arthrodesis and arthroplasty had a significantly higher revision rate. Did the authors perform statistical analysis for comparison of revision rate among the surgical techniques? In addition, which techniques were compared with arthrodesis and arthroplasty? Simple trapeziectomy? LRTI?	Apologies that this is not clearer - we did undertake analysis to compare the rate of revision between surgical subtypes, and the full model results are given in supplementary table 8 with the results shown in a forest plot in Figure 4. We then looked at the impact of sex within each surgical subgroup as sensitivity analyses, and those results in supplementary tables 9-12. We have added further explanation into the main text to show this, and to better signpost to the supplementary material.	Further explanation in results, subsection 'factors associated with further procedure'	Rewritten section factors associated with further procedure page 9

Reviewer: 2

Reviewer comments	Author response	Changes made	Page number in revised document
Main Concerns: 1. My main concern is the reliability and representativeness of the registration of complications and additional procedures using OPCS and ICD codes. Ergo, is the	This dataset enables us to identify the incidence of serious complications that require revision surgery or any form of admission to hospital including day case. This dataset covers a whole nation where the majority of healthcare is within a public system, and this public system records all payments for service. We believe that this research has something to offer the surgical community and patients by identifying these serious complications	1. changing the title to reflect that only serious complications are identified, and ensure this is clearer in the discussion and conclusions to emphasise that this study cannot identify ALL complications	Throughout all pages

reported number of complications and additional procedures reliable? My concern is that there may be a high risk of under reporting of complications , as we know that this is a frequent problem in studies on complications . Also, it may be that complications were not registered as requiring a new treatment episode. The reported numbers of complications seem unlikely low: e.g., in sup table 6 the numbers per surgery type are so low that it may be unrealistic. The numbers are much lower compared to other studies as well (e.g., Vermeulen et al. 2014 reports 25-37% complications). My doubts	discussed at the point of consent for surgery. We have worked on making it much clearer in this revised manuscript that this dataset has limitations - it cannot identify any complication that does not require surgery or admission. As it only looks at secondary care and is a national cohort of patients in routinely collected data, it is not comparable to the studies quoted here where further information can be gathered within a trial setting, but similarly, it reflects real-life practice across a whole nation rather than smaller trial populations. These are complimentary methodologies, and one should not exclude one kind of study. Whilst further information is gathered in a trial setting, there is also selection bias of patients of those who participate in research that is not seen when evaluating all routine healthcare work undertaken in on a national level. With regards to discussing the incidence of pre-existing medical conditions- this is a secondary care dataset, so will only identify patients who have needed hospital care for that condition. We acknowledge that as a limitation, but do not feel this prevents the study from identifying the incidence of serious complications, as it is extremely unlikely that serious complications would be treated anywhere else other than in secondary care in England within the NHS, and would be identified in this dataset.	2. Addition of further discussion of methodology in order to reassure about the steps taken to establish the accuracy of identification of complications that is well established in the literature 3. Additional emphasis in the introduction, aims, discussion and conclusions to show what this data can add to the literature but also its limitations, including that of the prevalence of pre-existing conditions 4. Reduction in emphasis within the abstract and discussion and conclusions so that readers do not infer causal inference when only associations can be made.	
---	---	--	--

are strengthened by the impression that the reported numbers in sup table 4 on comorbidity (e.g., CTS, Knee OA, etc.) are low, especially since the population of CMC-1 OA has a higher prevalence compared to their peers of similar age. It is unclear how the authors addressed this. Hence, I think there is a high risk of bias here and the authors cannot safely study complication or additional procedure rate using this data, nor draw this conclusion. My advise would be to only study the prediction of these events and not imply that these rates are reliable.				
2. In line with this, the classification	This paper set out to find only the most serious complications that present in secondary care- this study	We appreciate this. We have changed the	Change of term additional procedure to	throughout

of complications seems not really suitable. E.g., tendinitis, scar tenderness, temporary sensory loss, neuromas, thumb collapse etc., are not included here, whereas they should be considered a complication. Also, the time frame of 30 and 90 days is relatively short for these kind of joint procedures. Furthermore, complications that are considered systemic may not be correlated with the initial treatment, as, for example, stroke seems not related.	did not set out to discover other complications that would present in an outpatient setting or primary care - a randomised controlled trial is a much better design to look for these complications, and several have been published already, along with a Cochrane review. We have further developed the paper throughout to ensure that this is made clearer. We still feel that being able to counsel patients on the risk of serious adverse events is important for surgeons and their patients, and having this large national cohort of routine care in a nationalised system where there is a reduced risk of selection bias is the appropriate methodology to answer this question. As we are only looking at serious complications, the 30 and 90 day windows were undertaken for two reasons. Firstly, to align with the reporting for serious complications occurring after all surgeries that are reported on the NHS dashboard (https://digital.nhs.uk/data-and-information/publications/ci-hub/nhs-outcomes-framework.) to enable comparisons to be made with surgeries in other parts of orthopaedics (https://pubmed.ncbi.nlm.nih.gov/30262336/)	term to 'Additional thumb base procedure' as suggested.	additional thumb base procedure.	
7. In the statistics section (page 7 lines 10-16), the authors describe the use of 95% CI's to determine significance	Apologies that this was not made clearer- use of 95% CIs is used as a measure of identifying significance rather than the p value due to multiple tests occurring that means that p values are likely to be seen as 'significant' merely due to chance. Evaluating the hazard ratio, and the surrounding size and direction of the 95% confidence interval enables the magnitude and direction of impact to	Further detail added into the methods-statistical analysis section	Page 6 paragraph 4	

in light of multiple testing. This is unclear and does not seem logical to me in preventing a multiple testing problem. Please explain in more detail. Also, I think many readers will not be familiar with subhazard ratios, the authors may consider to explain this in more detail as well.	be assessed giving a more rounded picture than a single p-value. We appreciate that the Fine and Gray adjustment for the competing risk of mortality may not be well known, and therefore further information about this statistical test has been added to explain what a subhazard ratio is. However, we do not feel that exhaustive explanations of statistical techniques is appropriate to this article – an interested reader can find them in standard medical statistics textbooks.		
8. The classification of ethnicity seems inconsistent, and I doubt the need of reporting this.	Ethnicity is collected in a specified way by the NHS that we have used (https://digital.nhs.uk/data-and-information/data-tools-and-services/data-services/hospital-episode-statistics/hospital-episode-statistics-data-dictionary) We feel that it is pertinent to include ethnicity as the benefit of using a national dataset is to gain more information about patients of all ethnicities, and to gain greater insight into the role that ethnicity takes in disease incidence. Often, patients of non-white ethnicity are excluded from RCTs.	Further detail in methods regarding the reason for inclusion of ethnicity	Page 5, covariates paragraph
9. There is no referral to figure 2, I think this should be in the part on increasing trend of surgeries. Also, the description of	Apologies that the reference to figure 2 was omitted- this has been added, with further information in of how and why age specific incidence was calculated to link back to the methods section.	Further detail in statistical methods section, and results 'Trends in Surgery' section	Page 6 statistical methods paragraph 1 and Page 7, trends in surgery paragraph

this figure is quite unclear regarding the age-specific incidence.			
10. It seems illogical to include separate models per surgical procedure when there is also a model with surgical procedure as a factor. This may be a bit overwhelming .	Multivariable models were undertaken for all patients and then subsequently within each surgical group in order to give further detail for each of the surgical cohorts. Some believe that there is the potential for confounding by indication if all the surgical patients are grouped together in one analysis, and therefore these models specific to one surgical subtype enables a sensitivity analysis to be undertaken to address this concern. We feel that these supplementary models compliment the main analysis and would recommend them stay in, but only within the supplementary information for the interested reader. We feel this is important especially as the other reviewer has requested more information about surgical subtype and risk of further procedure.	Further explanation in the methods to explain why models for each surgical subtype were undertaken	Page 7, new paragraph 2
11. In the discussion, the authors conclude that surgery for CMC-1 OA is safe. I am not sure if that would be my conclusion based on this study. Also, Page 9 lines 48-51 and page 10 lines 3-6: see my main concern; I do not think that this study provides an accurate view of the number of complications	We appreciate that this can be seen as too assertive, however we have demonstrated the risk of serious complications is low. We have reworded the conclusions in order to dampen down the potential for causal assumption, and to reiterate that this study only investigates serious adverse events that require inpatient admission or surgery.	Rewritten discussion and conclusions to relate to serious complications only	Page 10 principle findings, extension of limitations section page 11, and amended abstract page

and additional procedures due a high risk of bias in registration.			
--	--	--	--

VERSION 2 – REVIEW

REVIEWER	Tsujii, Masaya Mie University
REVIEW RETURNED	14-Dec-2020

GENERAL COMMENTS	Dear authors, thank you for your revision of your manuscript. I think that the authors have addressed the comments of the reviewers appropriately.
--

REVIEWER	Wouters, Robbert Erasmus Universiteit Rotterdam
REVIEW RETURNED	17-Jan-2021

GENERAL COMMENTS	In this revised manuscript, the authors have addressed many relevant points very well and the manuscript has improved a lot. I think many important points are now fixed, however, some issues still remain. Although major points, I think these can be easily addressed, mostly by changing terminology etc. Please find my recommendations below: * Whilst the terminology is now better (i.e., addition of “serious” to complications), I still feel uncomfortable with the rate of complications, for example the ones mentioned per surgery type, (supp table 6). E.g., it seems very unlikely that in a total of >2000 arthrodeses less than 7, and in >1600 arthroplasties zero wound dehiscences and infections occurred. This is simply not credible. My suggestion would be that, throughout the manuscript, the authors would speak of registration of complications in stead, as registered complications is what is reported. In line with this, the authors may further nuance their conclusions and expand limitations. This is the most important issue to my opinion and needs to be addressed more adequately before this paper can be published. * Regarding time frame and systemic complications; this terminology remains strange, as I have never seen e.g., a myocardial infarction as e result of thumb base surgery. My suggestion would be to use the term systemic event, or something like that. Regarding time frame of 30 and 90 days; this is also probably for feasibility/convenience? (probably only registered for those time points?). Hence, perhaps mention this? * The authors have added to the manuscript that “Minimum follow up was set at one day”. To my opinion, this is really not logical and this seems very strange if complications are rated at 30 and 90 days. Hence, it would make sense to exclude patients without >90 days follow-up. As a clinician, I do not think it is logical that “further procedure or serious adverse events can occur very shortly after surgery (for example, acute carpal tunnel syndrome requiring decompression).” Most complications or readmissions will follow later. Minor;
---

	* In the previous review I commented on the authors' use of 95% CI's to prevent multiple testing. In the response to reviewer letter, the authors state that "95% confidence interval enables the magnitude and direction of impact to be assessed giving a more rounded picture than a single p-value". This is true of course, but to the best of my knowledge, this does not address a multiple testing problem. If one wants to address this using confidence intervals, one should enlarge the confidence interval (e.g., to 97.5% in the case of two primary outcomes). Reporting 95% CI's over p-values can be valuable to better indicate the variability but this does not address multiple testing (as Bonferroni does). My suggestion would be to not include the term multiple testing but to consider these analyses explanatory. * Despite the authors' explanation, I still think that the number of models (and associated number of (supplementary) tables) is a lot to process for many readers and may be overwhelming. My suggestion is still to omit and only use the main model.
--	--

VERSION 2 – AUTHOR RESPONSE

Point by point responses

Reviewer: 1- no issues raised. 'I think that the authors have addressed the comments of the reviewers appropriately'

Reviewer: 2

Reviewer comments	Author response	Changes made	Page number in revised document
Whilst the terminology is now better (i.e., addition of "serious" to complications), I still feel uncomfortable with the rate of complications, for example the ones mentioned per surgery type, (supp table 6). E.g., it seems very unlikely that in a total of >2000 arthrodeses less than 7, and in >1600 arthroplasties zero wound dehiscences and infections occurred. This is simply not credible. My suggestion would be that, throughout the manuscript, the authors would speak of registration of complications instead, as registered complications is what is reported. In line with this, the authors may further nuance their conclusions and expand limitations. This is the most important	The term 'registered' has now been added throughout the text in order to address the reviewers concern. We had used the word 'identified' in order to discuss that HES APC will only find complications requiring hospital admission as is discussed throughout, but have now changed this to address his concern.	The term 'registered' has been added throughout the text	Throughout

issue to my opinion and needs to be addressed more adequately before this paper can be published.	The rates given here are what this large national dataset shows, whether credible or not in the eyes of the reviewer. We have not manipulated the findings in any way, and our methodology has been refined through a programme of work using these techniques. A full discussion of the way that complications are defined (ie requiring a hospital admission) is already included in the paper, and we should leave readers to draw their own conclusions regarding this, not use the opinion of one reviewer to cloud their judgement.		
Regarding time frame and systemic complications; this terminology remains strange, as I have never seen e.g., a myocardial infarction as a result of thumb base surgery. My suggestion would be to use the term systemic event, or something like that. Regarding time frame of 30 and 90 days; this is also probably for feasibility/convenience? (probably only registered for those time points?). Hence, perhaps mention this?	The term 'event' had been added in order to address the reviewers concern regarding systemic complications. Undergoing general anaesthesia is a risk factor for systemic events and this is often discussed in surgical literature. As our study shows, they are rare and this is the great strength of a large national cohort as it can show the evidence of rare events that may not be seen at a local level. As the discussion of the role of WALANT and surgery under local anaesthetic increases in hand surgery, we included systemic complications in the	The term 'event' had been added throughout the text Additional clarification of reason for timeframes on page 6 paragraph 2	Throughout

	paper in order to evaluate the risks associated with a general anaesthetic. We maintain that this adds to the evidence and literature surrounding the discussion of hand surgery delivery. The timeframes of 30 and 90 days were included due to  1. clinical experience of the incidence of these events and 2. to enable comparison with other conditions both within the research literature and as part of the NHS quality outcomes framework that specifically records the number of events within 30 and 90 days. This was already discussed on page 6 paragraph 2, but further clarification has been added in order to aid understanding		
The authors have added to the manuscript that "Minimum follow up was set at one day". To my opinion, this is really not logical and this seems very strange if complications are rated at 30 and 90 days. Hence, it would make sense to exclude patients without >90 days follow-up. As a clinician, I do not think it is logical that "further procedure or serious adverse events can occur very shortly after surgery (for example, acute carpal tunnel syndrome requiring decompression)." Most complications or readmissions will follow later.	As a group of experienced hand surgeons, we can confirm that complications do indeed occur within 1 day of these operations. We would reiterate this is the reason for the follow up period beginning on day 1, and this is in order to prevent under reporting and selection bias by starting follow up later. We therefore feel that this apriori analysis plan has both clinical and	NA	NA

	epidemiological reasoning.		
Minor * In the previous review I commented on the authors' use of 95% CI's to prevent multiple testing. In the response to reviewer letter, the authors state that "95% confidence interval enables the magnitude and direction of impact to be assessed giving a more rounded picture than a single p-value". This is true of course, but to the best of my knowledge, this does not address a multiple testing problem. If one wants to address this using confidence intervals, one should enlarge the confidence interval (e.g., to 97.5% in the case of two primary outcomes). Reporting 95% CI's over p-values can be valuable to better indicate the variability but this does not address multiple testing (as Bonferroni does). My suggestion would be to not include the term multiple testing but to consider these analyses explanatory.	We agree regarding the term 'multiple testing' and have double checked the manuscript to ensure that this is not included. We appreciate this was written in the discussion with the reviewer, but as this is not included in the manuscript no changes have needed to be made to address this point.	NA	NA
Despite the authors' explanation, I still think that the number of models (and associated number of (supplementary) tables) is a lot to process for many readers and may be overwhelming. My suggestion is still to omit and only use the main model.	Providing all analyses within a supplement is something that we support as part of an open science ethos. It enables interested readers to review if they wish, and provides as much evidence as we can to those who wish to appraise our work. Whilst we appreciate your point, we feel that there is no reason not to publish full data for those readers who are interested as a supplementary file. To not do so is anti-scientific.